# Hierarchical Surface Structures and Large-Area Nanoscale Gratings in As_2_S_3_ and As_2_Se_3_ Films Irradiated with Femtosecond Laser Pulses

**DOI:** 10.3390/ma16134524

**Published:** 2023-06-22

**Authors:** Dmitrii Shuleiko, Stanislav Zabotnov, Olga Sokolovskaya, Maksim Poliakov, Lidiya Volkova, Tatiana Kunkel, Evgeny Kuzmin, Pavel Danilov, Sergey Kudryashov, Dmitrii Pepelayev, Sergey Kozyukhin, Leonid Golovan, Pavel Kashkarov

**Affiliations:** 1Faculty of Physics, Lomonosov Moscow State University, 1/2 Leninskie Gory, 119991 Moscow, Russia; shuleyko.dmitriy@physics.msu.ru (D.S.); zabotnov@physics.msu.ru (S.Z.); oi.sokolovskaja@physics.msu.ru (O.S.); kashkarov@physics.msu.ru (P.K.); 2Institute of Nanotechnology of Microelectronics of the Russian Academy of Sciences, 16A Nagatinskaya St., 119991 Moscow, Russia; maxsimpolykovv@gmail.com (M.P.); lidiya.volkova.96@mail.ru (L.V.); 3Moscow Institute of Physics and Technology, 9 Institutskiy Per., 141701 Dolgoprudny, Russia; kunkel.ts@phystech.edu; 4Lebedev Physical Institute, The Russian Academy of Science, 53 Leninsky Avenue, 119991 Moscow, Russia; kuzmine@lebedev.ru (E.K.); danilovpa@lebedev.ru (P.D.); kudryashovsi@lebedev.ru (S.K.); 5Institute of Advanced Materials and Technologies, National Research University of Electronic Technology, 1 Shokina Sq., 124498 Zelenograd, Russia; lpi@org.miet.ru; 6Kurnakov Institute of General and Inorganic Chemistry of the Russian Academy of Sciences, 31 Leninsky Avenue, 119991 Moscow, Russia; sergkoz@igic.ras.ru; 7National Research Centre “Kurchatov Institute”, 1 Akademika Kurchatova Sq., 123182 Moscow, Russia

**Keywords:** arsenic selenide, arsenic sulfide, femtosecond laser modification, hierarchical laser-induced periodic surface structures, surface plasmon polaritons

## Abstract

Chalcogenide vitreous semiconductors (ChVSs) find application in rewritable optical memory storage and optically switchable infrared photonic devices due to the possibility of fast and reversible phase transitions, as well as high refractive index and transmission in the near- and mid-infrared spectral range. Formed on such materials, laser-induced periodic surface structures (LIPSSs), open wide prospects for increasing information storage capacity and create polarization-sensitive optical elements of infrared photonics. In the present work, a possibility to produce LIPSSs under femtosecond laser irradiation (pulse duration 300 fs, wavelength 515 nm, repetition rate up to 2 kHz, pulse energy ranged 0.03 to 0.5 μJ) is demonstrated on a large (up to 5 × 5 mm^2^) area of arsenic sulfide (As_2_S_3_) and arsenic selenide (As_2_Se_3_) ChVS films. Scanning electron and atomic force microscopy revealed that LIPSSs with various periods (170–490 nm) and orientations can coexist within the same irradiated region as a hierarchical structure, resulting from the interference of various plasmon polariton modes generated under intense photoexcitation of nonequilibrium carriers within the film. The depth of the structures varied from 30 to 100 nm. The periods and orientations of the formed LIPSSs were numerically simulated using the Sipe–Drude approach. A good agreement of the calculations with the experimental data was achieved.

## 1. Introduction

Chalcogenide vitreous semiconductors (ChVSs) based on sulfide, selenide, and telluride alloys in binary or multicomponent systems are very promising materials for various optical and photonic applications in the spectral range from 0.6 to 20 µm [1,2]. Typically, ChVS-based bulk glasses or thin films are employed in various passive optical devices, e.g., lenses, windows, planar waveguides, etc. However, nowadays, these materials, due to their high optical nonlinearity [3], attract a great deal of interest as prospective media for fabricating active and nonlinear optical components, such as laser fiber amplifiers [4,5], and all-optical signal processing in near-infrared (IR) telecommunications [3]. Mid-IR sensing platforms for molecular tracking could also benefit from the development of these materials [3,6]. For both telecommunication and sensing applications, the current trend is minimizing device footprints and costs by implementing integrated optical components.

Femtosecond laser modification (FLM) of ChVSs opens up wide opportunities for their micro- and nanostructuring, resulting in the occurrence of their novel functional properties. FLM provides local and contactless light–matter interaction, which is remarkable for high velocity of phase transformations and precise surface treatment. Thus, significant structural modification, including formation of amorphous/crystalline areas in the bulk of the irradiated material, could be achieved, with the energy thresholds of crystallization and amorphization being reduced [7,8,9].

The phase transformation processes lie at the core of the writing and erasing information in ChVSs [10,11]. Since the exposure of femtosecond laser pulses is shorter than the characteristic times of thermal diffusion, it minimizes phase transitions time and thermal effects outside the irradiated area. Thus, the maximum possible locality and control over structural modification are additionally provided when the required phase state is realized in a clearly defined area corresponding to the focusing of the laser beam.

Another example of a promising FLM application is the formation of laser-induced periodic surface structures (LIPSSs), or ripples, with wave- and subwavelength periods on the surface of the thin film. Such surface texturing allows reflectance [12,13], absorption [14], band-gap width [15], wettability [16,17] and tribological properties [18,19] for different materials to be controlled. Additionally, LIPSSs are of special interest for optical sensorics owing to surface-enhanced Raman scattering [20,21] and for biomedicine since LIPSS-coated implants possess reduced bacterial adhesion [22,23].

In particular, LIPSSs have been actively studied for the well-known phase-change material Ge_2_Sb_2_Te_5_ (GST225). It was shown that the origin of the LIPSS formation is the generation of surface plasmon polaritons (SPPs) upon intense photoexcitation of free charge carriers [24,25]. These structures find their application as high-quality phase diffraction gratings consisting of alternating amorphous and crystalline stripes [26,27]. Pronounced optical and electrical in-plane anisotropy [28] of LIPSS could be further used as new photonics and nonvolatile memory elements that are sensitive to the direction of incident polarized light or applied current.

Due to the wide optical band gap (about 2 eV) of binary ChVSs such as arsenic sulfide (g-As_2_S_3_) and arsenic selenide (g-As_2_Se_3_), multiphoton absorption can be expected. This, in turn, could lead to the creation of LIPSSs with various parameters. The band gap for these compositions significantly exceeds the similar one for GST225 (2.32 eV for g-As_2_S_3_, 1.76 eV for g-As_2_Se_3_, and 0.76 eV for GST225 [29]); therefore, the use of binary compounds can significantly expand the transparency window and potentially reduce optical losses in LIPSS-based devices.

A wide variety of LIPSSs is well known: surface one-dimensional gratings, ring-shaped concentric structures, and nanopillars as a result of FLM for amorphous As_2_S_3_ thin films [30]. The type of structures, their period, and orientation are determined by the pulse number and laser irradiation fluence. However, different types of structures can be observed within the same irradiated area; as a result, the so-called hierarchical structures consisting of LIPSSs with different morphology and period can be formed.

Despite the fundamental and applied interest in the laser modification of As_2_Se_3_ thin films [31], there are no experimental studies on the LIPSS formation on this materials.

Our study presents experimentally obtained various LIPSS types for amorphous As_2_S_3_ and As_2_Se_3_ thin films, including the hierarchical structures upon femtosecond laser irradiation with a different number of pulses with different energy. In addition to the experiment, LIPSS formation was also simulated within the framework of the Sipe–Drude theory [32,33,34].

## 2. Methods

### 2.1. Experimental Samples and Methods

The ChVS films were prepared by thermal evaporation of the synthesized glasses of As_2_S_3_ and As_2_Se_3_ in vacuum chambers. Residual pressure in the chamber was 10^−4^ Pa. The films were deposited on borosilicate glass substrates (Corning 1737F Glass, Corning, Corning, USA) and Cr (20 nm)/SiO_2_ (1 μm)/Si (substrate) multilayer structures (Figure 1a). The chromium layer was deposited to improve adhesion of the ChVS layer. In turn, the dielectric silicon oxide sublayer minimizes influence of photoexcited free charge carriers from the silicon substrate on LIPSS formation exactly in the ChVS layer. The thickness of the As_2_S_3_ and As_2_Se_3_ films measured by the atomic force microscopy (AFM, Solver P47-Pro, NT-MDT, Moscow, Russia) amounted to 576 ± 5 and 842 ± 5 nm, respectively. The elemental composition of the initial films was evaluated using scanning electron microscope (SEM JEOL JSM-6010, Akishima, Japan) equipped with an energy-dispersive spectrometer (JEOL EX94400T4L11 Akishima, Japan).

Irradiation was carried out in air at normal incidence by a focused femtosecond laser beam (Satsuma, Amplitude Systems, Pessac, France) at the frequency of the second optical harmonic (λ = 515 nm, τ = 300 fs). The spatial intensity distribution in the beam was Gaussian. The polarization of the beam was linear. To determine optimal regimes of the LIPSS formation, at the first stage, two-dimensional arrays of craters were formed at the surfaces of each of the films by the laser beam focused to the spot of 15 μm in diameter. Within the formed arrays, the number of laser pulses N increased from 10 to 1200 along one dimension, and the energy E per pulse varied from 0.03 to 0.5 μJ. The laser pulse fluence *F* varied from 17 to 270 mJ/cm^2^, respectively. The production of the crater arrays was carried out with very low repetition rate to ensure the precise control of the laser pulse number. For N below 100 the repetition rate was 1Hz, whereas for N above 100 the repetition rate was 10 Hz.

At the second stage, based on the results of the morphological analysis of LIPSS in the formed crater arrays, the irradiation parameters were chosen to create a large-area LIPSS structures in As_2_Se_3_ thin film. The irradiation of large (compared to the LIPPS period and laser beam diameter) areas (5 × 5 mm^2^) was carried out in the scanning mode, which was realized via moving the sample by a system of 2 automated mechanical translators. The laser spot moved along one axis continuously at a speed *v* = 300 μm/s, and discretely in the orthogonal direction with a step *d* equal to the half of the laser spot diameter *D* (Figure 1b). Two samples were formed at the same laser radiation fluence *F* = 50 mJ/cm^2^ but with different pulse repetition rates, 1 and 2 kHz. As a result, the number of pulses per unit area *N* was equal to 400 and 800, respectively. An example of the formed sample is given in Figure 1c.

The morphological features of the irradiated surfaces of the As_2_S_3_ and As_2_Se_3_ films were studied by SEM (MIRA, Tescan, Czech Republic; Helios G4CX FEI, Thermo Fisher Scientific Inc., Waltham, MA, USA) and AFM (NEXT II, NT-MDT, Moscow, Russia). The LIPSS periods were found as the average of 10 or more periods measured at an analysis of AFM profiles.

The chemical composition of initial and femtosecond laser-irradiated As_2_S_3_ film was analyzed by the energy-dispersive X-ray spectroscopy method (EDX), using a Tescan Vega 3 SEM with an Oxford Instruments Xplore energy-dispersive analyzer.

The fine structure and chemical composition, including the profile distributions of the chemical elements across the film thickness, for the As_2_Se_3_ film irradiated in scanning mode by femtosecond laser pulses, were studied by means of transmission electron microscopy (TEM, JEOL JEM 2100 Plus, JEOL Ltd., Akishima, Japan) equipped with a JEOL EX-24261M1G5T energy-dispersive analyzer. TEM cross-section specimens were prepared using dual-beam workstation Helios G4CX FEI with micromanipulator by the in situ lift-out method.

### 2.2. Numerical Simulation of the Formation of Hierarchical Structures

The Sipe–Drude model described in Refs. [24,32,33,34] was used to simulate the probability of the LIPPS formation with the reduced wave vectors *k*_x_ and *k*_y_ values in the sample surface plane depending on the parameters of the acting laser radiation and the complex permittivity of the surface during irradiation. In this case, the values of the wave vectors *k*_x_ and *k*_y_ in the *xy* plane of the thin film are defined as
*k*_*x*,*y*_ = *λ*/*Λ*_*x*,*y*_,(1)
where *λ* is the wavelength of the acting laser radiation, and *Λ*_x,y_ is the LIPSS period.

This model allows calculating the two-dimensional distribution of the efficacy factor of surface gratings formation *η*(*k*_x_, *k*_y_), whose local maxima correspond to the most probable values of the wave vectors of the excited SPP, and, accordingly, the formed LIPSS. Both the parameters of the acting laser radiation (wavelength, angle of incidence, polarization) and the properties of the irradiated surface (surface roughness, complex permittivity) were taken into account in the calculations, including the excitation of nonequilibrium charge carriers with *N*_e_ concentration in the irradiated film by laser pulses. Excitation of these carriers changes the dielectric constant of the irradiated material according to the Drude model during the exposition with the laser pulse.

Numerical calculation of *η*(*k*_x_, *k*_y_), was carried out in the software package Wolfram Mathematica v11.0.0 using a system of equations for s-polarized radiation at normal incidence [34]. The values of the real part of the permittivity *ε* for As_2_S_3_ and As_2_Se_3_ were 7.7 and 11.5 according to the data from [35,36]. The wavelength of the incident laser radiation was *λ* = 515 nm, and the shape and filling factors determining the surface roughness were 0.4 and 0.1, respectively [34].

The concentration of nonequilibrium charge carriers *N*_e_ achieved in the irradiated film was estimated using the following differential equation:(2)$$\frac{dN_{e}}{dt}=\frac{\left(1-R\right)I(t)}{\hbar \omega }\alpha +\frac{{\left(1-R\right)}^{2}I^{2}(t)}{2\hbar \omega }\beta ,$$
where *α* and *β* are the one- and two-photon absorption coefficients, *R* is the reflection coefficient, and *I*(*t*) is the laser radiation intensity, calculated similarly to the method presented in [37]. The values of the coefficients of one- and two-photon absorption for the As_2_S_3_ film were taken equal to *α* = 1.3 × 10^6^ cm^−1^ [38] and *β* = 0.3 cm/GW [39,40], and for the As_2_Se_3_ film, *α* = 2.5 × 10^4^ cm^−1^ [38] and *β* = 0.14 cm/GW [39], respectively.

## 3. Results and Discussion

### 3.1. Surface Analysis of Irradiated Films

SEM images of crater arrays formed at different energies and numbers of laser pulses are shown in Figure 2. It can be seen that the threshold pulse energy, at which the modification of the film surface begins, is 0.36 μJ for As_2_S_3_ (*F* = 136 mJ/cm^2^) and 0.05 μJ for As_2_Se_3_ (*F* = 34 mJ/cm^2^), respectively. It is worth noting that for As_2_S_3_ the fluence exceeds the ablation threshold, which ranges from 7.2 mJ/cm^2^ to 24 mJ/cm^2^ for femtosecond 800 nm laser pulses according to Refs [41,42]. Presumably, modification of As_2_S_3_ requires higher energies of laser pulses because the As–S chemical bonds are stronger than those of As–Se (260 and 230 kJ/mol, respectively) [43]. Typically, ablation threshold for As_2_Se_3_ was reported to be almost twice less than for As_2_S_3_ [44,45].

When the energy of the laser pulse is less than the threshold for the As_2_S_3_ film modification, the SEM images did not reveal any modification. Starting from the laser pulse energy *E* = 0.36 μJ and the number of pulses *N* = 50, the formation of one-dimensional gratings is observed on the film surface, with grooves oriented along the laser radiation polarization (Figure 3a). These structures have a mean period of 200 nm, which is much less than the wavelength of the modifying laser radiation, and represent periodic gratings with the high spatial frequency (“high spatial frequency LIPSS”, or HSFL) [33]. With an increase in the number of pulses to 400 and higher, as well as at laser pulse energies above the threshold (0.36 μJ and higher), one-dimensional gratings with the low spatial frequency (“low spatial frequency LIPSS”, or LSFL) clearly appear in the center of the crater. The period of LSFL is ~430 nm and close to the wavelength of laser radiation. The orientation of the LSFL ridges is orthogonal to the polarization of the laser beam (Figure 3b,c). At *E* = 0.36 μJ, LSFL-type structures are observed in the center of craters starting from *N* = 10 (Figure 3c) and up to complete ablation of the films. Craters containing only HSFL or LSFL structures are indicated in Figure 2 with circles and squares, respectively.

At *E* = 0.24 μJ, in the range of the number of laser pulses from 100 to 400, both types of LIPSS are formed simultaneously in the crater, in the form of a hierarchical structure. In this case, with an increase in the number of pulses, a gradual destruction of the HSFL-type LIPSS in the center of the crater occurred with the simultaneous formation of an LSFL grating. An example of a hierarchical structure on the As_2_S_3_ surface is given in Figure 3b. Craters containing hierarchical structures are indicated in Figure 2 by rectangles with rounded edges.

In the case of an As_2_Se_3_ film, modification began at a lower threshold pulse energy (*E* = 0.05 μJ) and the same number of laser pulses *N* = 50, with the formation of similar one-dimensional HSFL-type structures with a mean period of 167 nm, oriented along the polarization of structuring pulses. At *E* > 0. 05 μJ, the formation of LSFL type structures is observed on the As_2_Se_3_ surface for all values of *N*, starting from 10, and up to complete ablation of the film in the center of the crater. The mean period of these LIPSS is 490 nm. The examples of pronounced HSFL and LSFL structures on the As_2_Se_3_ surface are shown in Figure 3d,f, respectively.

The formation of a hierarchical structure on the As_2_Se_3_ film surface at *E* = 0.05 μJ was not observed up to the number of laser pulses *N* = 800, when the generation of an LSFL-type LIPSS over the HSFL structure began in the central region of the crater. In this case, in contrast to the As_2_S_3_ film, the destruction of the fine-periodic HSFL structure during the appearance of a long-period LSFL grating in the center of the crater was not observed even at the maximum number of laser pulses used in our experiments, *N* = 1200, and the energy *E* = 0.05 μJ (Figure 3e). Nevertheless, similar to the As_2_S_3_ film, it can be seen that with higher fluence of laser pulses, only LSFL-type LIPSS remain in the center of the crater (Figure 3f).

The morphology of hierarchical structures on the surfaces of both film material types can be characterized by the presence of the long-period LSFL-type LIPSS at the center of craters and HSFL structures at the periphery (Figure 3b,e). Simultaneous presence of LIPSS of different types in different parts of the crater surface is explained by Gaussian intensity distribution in the laser beam. As a result, at any laser pulse energy above the threshold value, there always exists a region in the beam cross section where the radiation fluence corresponds to the HSFL structures formation.

According to AFM data, the modulation depth of HSFL type gratings on As_2_Se_3_ films reached 40 ± 3 nm at irradiation parameters *N* = 1200, *E* = 0.05 μJ (Figure 4a), averaging about 25 nm in other craters (Table 1). The modulation depth of the LSFL-type LIPSS was higher, with a maximum value of 115 ± 5 nm for *N* = 50, *E* = 0.12 μJ (Figure 4b), and average depth of 80 nm for other craters (Table 1). In the case of As_2_S_3_ films, the surface relief was highly inhomogeneous due to the presence of ablation products deposited on the irradiated surface, and the mean depth of HSFL and LSFL gratings modulation was approximately two times smaller than that for the As_2_Se_3_ film (Table 1). The depth of the crater coincides with the depth of the LIPSS.

Our study revealed no effect of the laser pulse repetition rate on the type of created surface structures for the repletion rate below 1 MHz, when the changes introduced by the previous laser pulse have not been relaxed by the time of irradiation by the next one.

At the maximum energy of laser pulses (0.48 μJ), the whole ablation of both types of films was observed in the center of craters, starting from the minimum number of pulses used (*N* = 10).

EDX analysis of structures did not reveal any change in the composition of the substance compared with the initial one. The EDX results prove the uniform distribution of elements across the as-deposited and irradiated films (Appendix A).

Since As_2_Se_3_ film requires less energy for its modification, the film of this composition was chosen for the large-scale periodic nanograting recording. Two sets of laser irradiation parameters were selected: the number of pulses per unit area *N* = 400 at fluence *F* = 50 mJ/cm^2^, and *N* = 800 at the same fluence. These two sets correspond to the transition from HSFL to hierarchical structure formation.

Two 5 × 5 mm^2^ square areas were formed on As_2_Se_3_ film surface using parameters mentioned above. SEM analysis revealed that at *N* = 400, only HSFL grating with 200 ± 10 nm period is formed on the film surface, without any hierarchical structure (Figure 5a). However, at N = 800, a hierarchical structure is observed (Figure 5b) in the form of a highly ordered two-dimensional grating. The periodicity along one axis is provided by the presence of LSFL, and in the orthogonal direction—by HSFL. Herewith, the ridges of HSFL orthogonally intersecting the LSFL grating exist not only in the valleys of the LSFL structure, but also on its ridges, as can be seen in Figure 5c. The LSFL and HSFL periods in this hierarchical structure were 490 ± 5 nm and 190 ± 10 nm, respectively. Thus, the irradiation parameters and the structures formed on the As_2_Se_3_ by femtosecond laser pulses in the scanning mode are consistent with the results obtained previously in the single craters formation mode.

The formed hierarchical surface relief depth was 60 nm according to the TEM analysis of the film cross section (Figure 5d). Additionally, TEM demonstrated that the As_2_Se_3_ films modified on the large scale remained amorphous: no phase transformations were observed in the irradiated areas. The depth of the formed structure is the same as LIPSS depth (Figure 5d).

Herein, it should be emphasized that the observed fabrication of the hierarchical LIPSSs does not require laser beam splitting for relief modification [46,47,48], i.e., using the direct laser interference pattering technique [49] or changing a scanning strategy [50]. Thus, we implemented a single-stage, single-beam, and scalable technique to form hierarchical LIPSSs in As_2_Se_3_ thin films. Moreover, to our knowledge, we formed LIPSSs on As_2_Se_3_ films for the first time.

### 3.2. Theoretical Explanation of LIPSS Formation

The formation of LIPSSs of various types upon varying the number and energy of laser pulses is explained by the generation of various SPP modes [32,37] on the surface of films, which becomes possible for ChVSs due to the intense photoexcitation of free charge carriers under the action of laser radiation with a high fluence [26]. Depending on the absorbed energy of laser radiation, the concentration of free carriers *N*_e_ changes, which affects the value of the complex dielectric permittivity during irradiation and, accordingly, the wave vector of the generated SPP (*k*_x_, *k*_y_), which determines the period of the emerging LIPSS.

Simulation within the framework of the Sipe–Drude theory allows calculating the periods and orientations of the LIPSSs and revealing the relationship between photoinduced charge carriers’ concentration and the type of structures formed.

For As_2_Se_3_ films, the results are given in Figure 6: the calculations of two-dimensional efficacy factor distributions *η*(*k*_x_, *k*_y_) as probabilities of SPP excitation with the corresponding wave vector for different irradiation conditions (right column), and their comparison with the Fourier transforms (middle column) of the corresponding AFM images of crater surfaces (left column). In all cases, the direction of the laser radiation polarization in calculations was parallel to the *x* axis.

It should be noted that the vertical and horizontal lines passing through Fourier image centers are not a consequence of the LIPSS formation, but are a result of the presence of artifacts associated with the discreteness of scanning samples by the AFM method. Additionally, in the case of As_2_S_3_ films, large fragments and ablation products deposited on the surface of the films make it difficult to scan the surface by AFM and analyze the obtained crater images. Therefore, for As_2_S_3_, the results are presented only as a comparison of the positions of *η*(*k*_x_, *k*_y_) local maxima and the experimentally obtained LIPSS periods.

For both types of films, two groups of local maxima *η*(*k*_x_, *k*_y_) were observed in the simulation. In the case of As_2_Se_3_ films, HSFL correspond to local maxima of the efficiency factor at *k*_x_ = 0, *k*_y_ ≈ 3.15 (Figure 6a), while LSFL correspond to local maxima at *k*_x_ ≈ 1.07, *k*_y_ = 0 (Figure 6c), which appear on the two-dimensional efficacy factor *η* distributions as large pairs of arcs above and below, or small pairs of arcs to the left and right of the center, respectively. In this case, the calculated LIPSS periods according to Formula (1) are 163 nm for HSFL and 481 nm for LSFL, which is close to the experimental results (Table 1). The ratio of the amplitudes of these local maxima changes with varying the concentration of photoinduced charge carriers *N*_e_, as can be seen in Figure 6a–c, which show the results of calculations corresponding to *N*_e_ values of 6.0 × 10^21^, 1.5 10^22^, and 2.9 × 10^22^ cm^−3^, respectively.

According to calculation results, the probability of HSFL formation remains higher than the probability of LSFL formation at excited nonequilibrium charge carriers concentration *N*_e_ < 1.5 × 10^22^ cm^−3^. As *N*_e_ increases, the local maxima of *η* corresponding to HSFL begin to shift towards smaller *k*_y_, blur, and completely disappear at *N*_e_ > 3 × 10^22^ cm^−3^, which corresponds to the formation of only LSFL-type gratings.

The values of *N*_e_ = 6.0 × 10^21^ and *N*e = 3.0 × 10^22^ cm^−3^ required for the formation of HSFL and LSFL are achieved at fluences of the laser radiation *F* = 40 mJ/cm^2^ and *F* = 70 mJ/cm^2^, respectively, according to Equation (2). These values of *F* agree with the experimental laser pulse energies of 0.05 and 0.12 μJ, at which the formation of the corresponding LIPSS was observed.

Similar results were obtained for As_2_S_3_, where the required *η* distributions corresponding to generation of HSFL and LSFL with the periods obtained in experiments are achieved at higher values of *N*_e_ = 7.5 × 10^21^ and 4.0 × 10^22^ cm^−3^ than ones for As_2_Se_3_, which, among other things, is related to different value of permittivity *ε* for the given material. According to calculations using Equation (2) and taking into account the values of the one- and two-photon absorption coefficients for As_2_S_3_, these values of *N*_e_ are achieved at fluence of the acting laser radiation *F* > 140 mJ/cm^2^, which is in line with experimental value *E* = 0.24 μJ.

The higher laser pulse energy required for the plasmon polariton excitation in the As_2_S_3_ is also likely a reason for the formation of LIPSS with less pronounced relief on the surface of these films, as the high fluence of laser radiation leads to more intense ablation of the film surface, destruction of high LIPSS ridges, and formation of a larger number of fragments.

## 4. Conclusions

Thus, we experimentally demonstrated the possibility of forming LIPSSs in As_2_S_3_ and As_2_Se_3_ thin films. These structures are gratings with different orientations and periods: the HSFL structure with subwavelength period and ridges parallel to the polarization of femtosecond laser pulses; and the LSFL structure with period close to the wavelength of laser radiation directed orthogonal to the laser polarization. The periods and orientations of the structures depend on the fluence and number of laser pulses. Herewith, in a certain range of energy and laser pulses number, both LIPSS types are generated simultaneously in the crater as a hierarchical structure.

The formation of periodic structures is explained by the photoinduced generation of free charge carriers in a thin surface layer of a semiconductor, which leads to the appearance of SPP. The periods of the obtained LIPSSs are in a good agreement with the results of simulation within the Sipe–Drude theory, which demonstrates, among other things, the dependence of the same LIPSS type period on the value of the initial films dielectric permittivity.

The possibility of scaling up to 5 × 5 mm^2^ areas covered by various LIPSS types, including hierarchical two-dimensional gratings, was shown via femtosecond laser irradiation of the As_2_Se_3_ film in the scanning mode using only one laser beam. Such 2D periodical gratings on the irradiated As_2_S_3_ and As_2_Se_3_ surfaces may cause optical artificial anisotropy and are of interest for designing elements of polarization optics and photonic integrated circuits for the infrared range.

## Figures and Tables

**Figure 1 materials-16-04524-f001:**
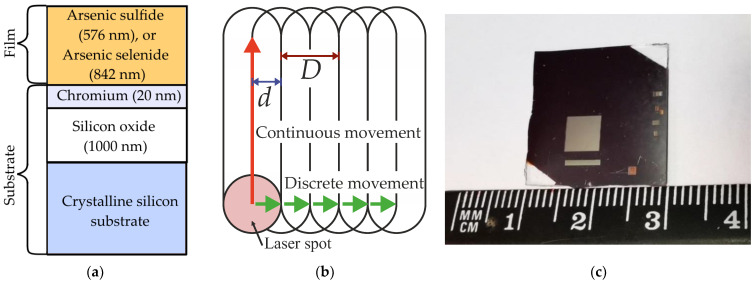
(**a**) Scheme of the samples deposited on the multilayer substrate. (**b**) Scheme of the laser spot movement in scanning mode. (**c**) Photo of the irradiated area on the As_2_Se_3_ film surface.

**Figure 2 materials-16-04524-f002:**
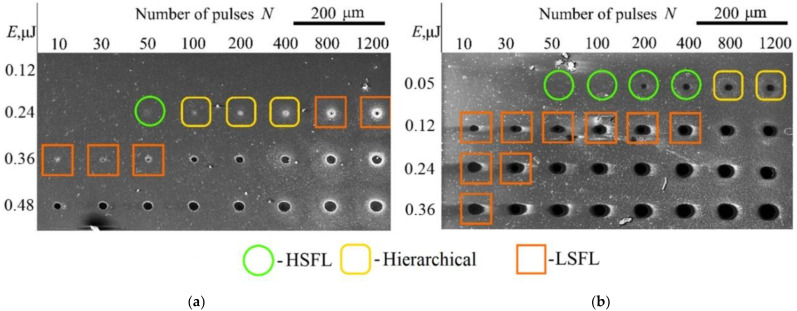
SEM images of arrays of craters on the surface of (**a**) As_2_S_3_ and (**b**) As_2_Se_3_ films after irradiation with a different number of femtosecond laser pulses with different energies. Frames of various shapes encircle craters in which the formation of structures of the HSFL, LSFL type, as well as hierarchical structures (HSFL + LSFL) was observed. Polarization of laser radiation is horizontal.

**Figure 3 materials-16-04524-f003:**
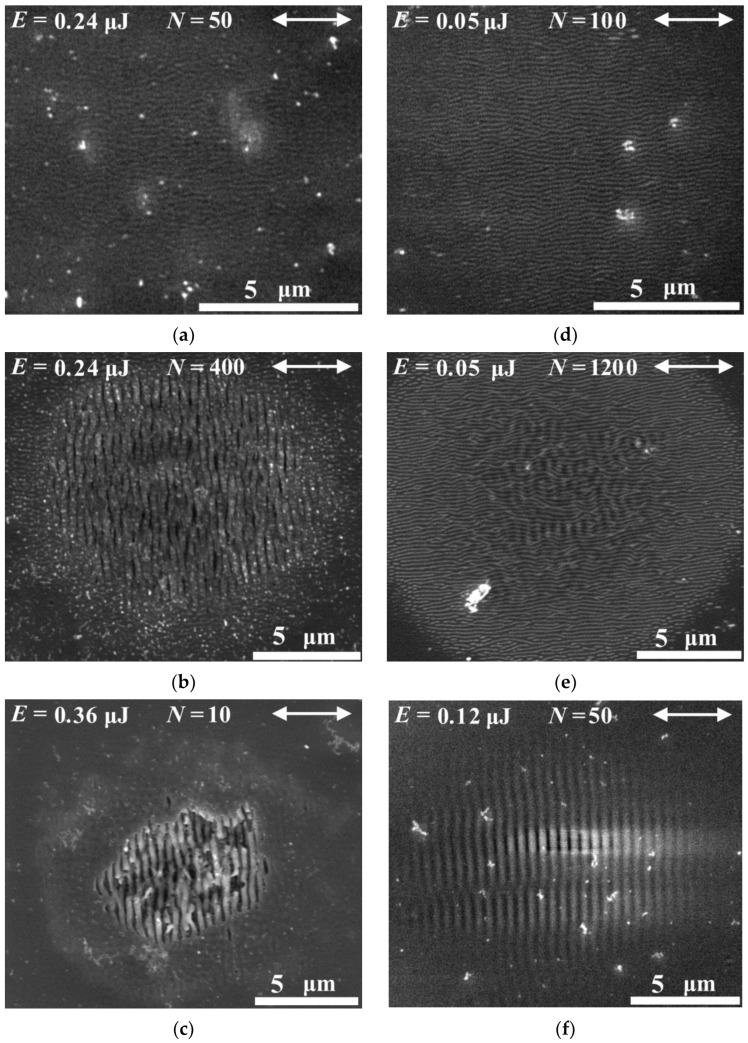
SEM images of LIPSS of various types on the films (**a**–**c**) As_2_S_3_ and (**d**–**f**) As_2_Se_3_, obtained by irradiation with femtosecond laser pulses. The irradiation parameters are indicated in the images, the laser radiation polarization, indicated in all images by an arrow, is horizontal.

**Figure 4 materials-16-04524-f004:**
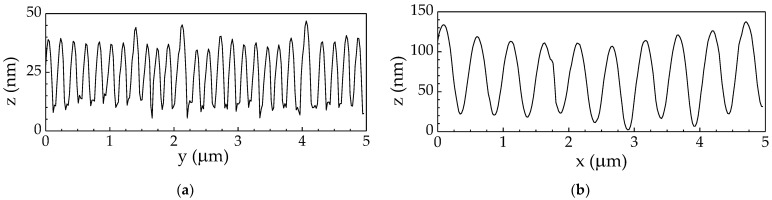
AFM profiles at the maximal depth modulation of the craters with LIPSS of (**a**) HSFL (*N* = 1200, *E* = 0.05 μJ) and (**b**) LSFL (*N* = 50, *E* = 0.12 μJ) types in the As_2_Se_3_ films. Here, z is height, and x and y are coordinates parallel and perpendicular to the laser polarization, correspondingly.

**Figure 5 materials-16-04524-f005:**
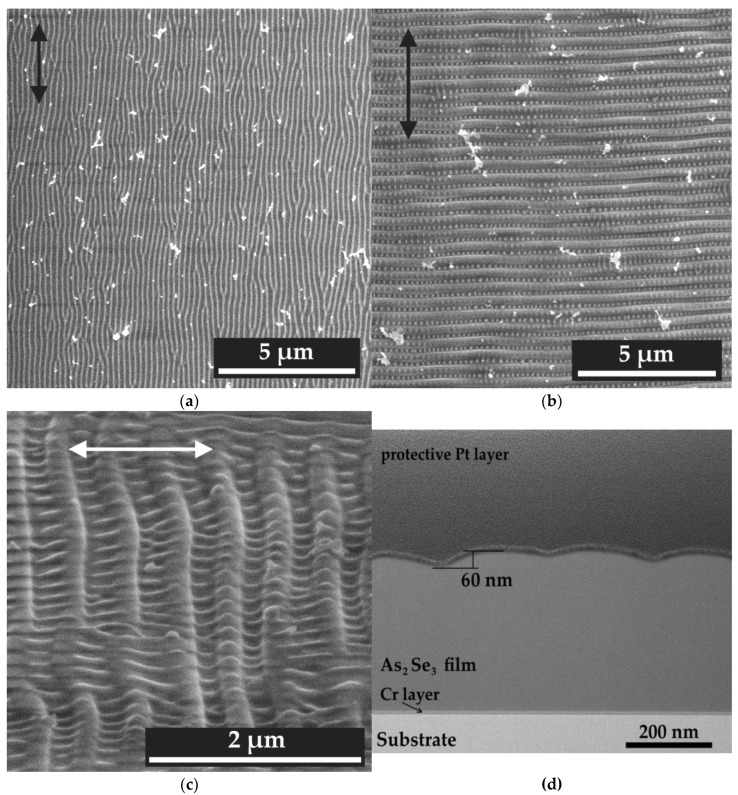
SEM images of the nanoscale periodic gratings obtained on the As_2_Se_3_ film surface using femtosecond laser pulses at (**a**) 1 kHz and (**b**,**c**) 2 kHz repetition rate. Image (**c**) obtained at 52° tilt angle. The arrow indicates the polarization direction of femtosecond laser pulses. (**d**) TEM image of the cross section of periodic grating on As_2_Se_3_ film obtained by irradiation with femtosecond laser pulses at 2 kHz repetition rate.

**Figure 6 materials-16-04524-f006:**
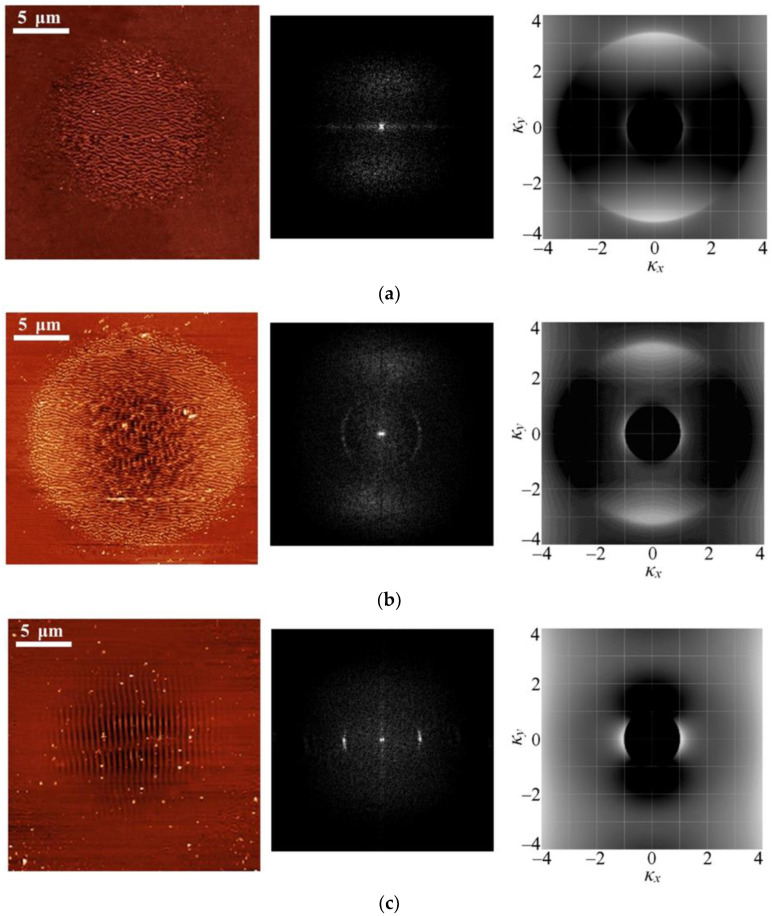
From left to right: AFM images of craters on the surface of As_2_Se_3_ films, Fourier transforms of image data and calculated probability of occurrence of LIPSS with the wave vector (*k*_x_, *k*_y_) normalized to the wavelength laser pulses under the following irradiation conditions: (**a**) 50 pulses with energy 0.05 μJ, (**b**) 1200 pulses with an energy of 0.05 μJ, and (**c**) 50 pulses with an energy of 0.12 µJ. The radiation polarization is horizontal.

**Table 1 materials-16-04524-t001:** Periods and mean depths of structures of various types obtained on As_2_S_3_ and As_2_Se_3_ films by femtosecond laser pulses.

	As_2_S_3_		As_2_Se_3_	
	Period, nm	Depth, nm	Period, nm	Depth, nm
HSFL	200 ± 10	13 ± 3	170 ± 10	25 ± 3
LSFL	430 ± 20	50 ± 20	490 ± 10	80 ± 10

## Data Availability

Not applicable.

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
