# Peer review of "Hierarchical Surface Structures and Large-Area Nanoscale Gratings in As2S3 and As2Se3 Films Irradiated with Femtosecond Laser Pulses"

_materials, 2023, doi:10.3390/ma16134524_

Round 1
Reviewer 1 Report
The manuscript by Shuleiko et al. reports on the morphological characterization of Laser Induced Periodic Surface Structures (LIPSS) on Chalcogenide vitreous thin films, and in particular, arsenic sulfide and arsenic selenide.
The manuscript contains a systematic study on the LIPSS morphological dimension using Scanning Electron Microscopy and Atomic Force Microscopy. LIPSS periodicity was evaluated as laser fluence and number of pulses changed. The manuscript also presents a theoretical analysis of LIPSS formation and suggests that it can be explained by the photoinduced generation of free charge carriers in the thin films which leads to the appearance of Surface Plasmon Polaritons.
General comments
The systematic study of LIPSS morphology for chalcogenide vitreous thin films as a function of pulse energy and pulse numbers is a strong point of the submission and the accompanying theoretical section is appealing as well. The manuscript is written in a clear way, using a good English. The work is interesting and will be appreciated by the scientific community. In fact, the topic of LIPSSs is still of great interest and the work of Shuleiko et al. presents an easy replicable method for the fabrication of said structures on a technological interesting class of materials.
The references presented in the work are appropriate and are used wisely to strengthen the scientific validity of the obtained result, however background references in the introduction are lacking.
The work is very interesting and provides elements of novelty, therefore I suggest publication only after addressing some issues that were spotted during the peer review process.
Specific comments
1. In the abstract the authors provide a direct measurement of the LIPSS periodicities, but the laser wavelength is not mentioned. Laser wavelength, along with other parameters such as laser repetition rate, pulse width and pulse energy (a range) should be mentioned in the abstract.
2. The authors mention in the manuscript that novel functional properties could be induced by the generation of LIPSS. However, no example or reference of that is reported. The recent literature is rich with wide bandgap semiconductor’s properties alterations, such as: optical alterations (reflectance, absorptance...), mechanical (friction, etc…) and even electronical (bandgap engineering). The authors should enrich this paragraph with more references.
3. Why is it used a multi-layer substrate?
4. In section 2 (methods), the authors do not disclose the laser repetition rate used for the fabrication of LIPSS in the array of “craters”. It is however reported that 1KHz and 2KHz were used for the fabrication of lines.
5. Can the repetition rate have an effect on LIPSS morphology?
6. Row 117: it is not entirely clear to what energy losses are related to.
7. Figure 1: is it the movement of the laser a raster scan?
8. Laser polarization should be indicated in all SEM images with arrows as it was done for figure 5.
9. Have the authors measured the depth of the produced “craters”? And of the produced “lines”? Was it considered when producing the large area samples?
10. How were the LIPSS periodicities measured? I imagine it was carried out by analyzing Fourier transform images, but it was not mentioned in the paper.
11. One of the strong points of the papers is that the authors were able to obtain two-dimensional LIPSS with a single processing step, a linear polarization and without using beam shaping or other techniques such as “photon doubling”. I suggest highlighting this and to compare it with recently employed techniques such as:
a. temporally delayed and cross-polarized linear pulses: https://doi.org/10.1021/acs.nanolett.1c01310
b. Circular polarization: https://doi.org/10.1016/j.apsusc.2018.11.106
c. direct laser interference patterning: https://doi.org/10.1103/PhysRevB.103.054105
The mentioned papers could give some context to the reader in the difficulty of obtaining such patterns.
Minor errors such as row 29 and 56.
Author Response
Response to Reviewer 1 Comments
We thank the reviewer for his substantial comments on our manuscript.
We have improved the manuscript text in accordance with the comments.
Comment 1: In the abstract the authors provide a direct measurement of the LIPSS periodicities, but the laser wavelength is not mentioned. Laser wavelength, along with other parameters such as laser repetition rate, pulse width and pulse energy (a range) should be mentioned in the abstract.
Response 1:
Thank you for your comment. We added mentioned information into the abstact:
In present work, a possibility to produce LIPSS under irradiation by 300 fs laser pulses (wavelength 515 nm, repetition rate up to 2 kHz, pulse energy ranged 0.03 to 0.5 μJ) is demonstrated on a large (up to 5x5 mm2) area of arsenic sulfide (As2S3) and arsenic selenide (As2Se3) ChVS films.
(lines 29-31)
Comment 2: The authors mention in the manuscript that novel functional properties could be induced by the generation of LIPSS. However, no example or reference of that is reported. The recent literature is rich with wide bandgap semiconductor’s properties alterations, such as: optical alterations (reflectance, absorptance...), mechanical (friction, etc…) and even electronical (bandgap engineering). The authors should enrich this paragraph with more references.
Response 2:
Thank you for your comment. We completely agree with it. We briefly discussed variations of material properties by means of LIPSS formation. The following text was added:
Such surface texturing allows reflectance [12,13], absorption [14], band-gap width [15], wettability [16,17] and tribological properties [18,19] for different materials to be controlled. Additionally, LIPSSs are of special interest for optical sensorics owing to surface-enhanced Raman scattering [20,21] and for biomedicine since LIPSS-coated implants possess reduced bacterial adhesion [22,23].
In particular, LIPSSs have been actively studied for the well-known phase-change material Ge2Sb2Te5 (GST225).
(lines 70-76)
with added references:
- Ou, Z.; Huang, M.; Zhao, F. The fluence threshold of femtosecond laser blackening of metals: The effect of laser-induced ripples. Las. Technol. 2016, 79, 79–87. DOI: 10.1016/j.optlastec.2015.11.018.
- Vorobyev, A.Y.; Guo, C. Antireflection effect of femtosecond laser-induced periodic surface structures on silicon. Express 2011, 19, A1031-A1036. DOI: 10.1364/oe.19.0a1031.
- Yang, Y.; Yang, J.; Liang, C.; Wan, H. Ultra-broadband enhanced absorption of metal surfaces structured by femtosecond laser pulses. Express 2006, 16, 11259–1265. DOI: 10.1364/oe.16.011259.
- Crouch, C.H.; Carey, J.E.; Shen, M.E.; Mazur, E.; Génin, F.Y. Infrared absorption by sulfur-doped silicon formed by femtosecond laser irradiation. Phys. A 2004, 79, 1635–1641. DOI: 10.1007/s00339-004-2676-0.
- Skoulas, E.; Manousaki, A.; Fotakis, C.; Stratakis, E. Biomimetic surface structuring using cylindrical vector femtosecond laser beams. Rep. 2017, 7, 45114. DOI: 10.1038/srep45114.
- Žemaitis, A.; Mimidis, A.; Papadopoulos, A.; Gečys, P.; Račiukaitis, G.; Stratakis, E.; Gedvilas, M. Controlling the wettability of stainless steel from highly-hydrophilic to super-hydrophobic by femtosecond laser-induced ripples and nanospikes. RSC Adv. 2020, 10, 37956–37961. DOI: 10.1039/d0ra05665k.
- Bonse, J.; Koter, R.; Hartelt M.; Spaltmann D.; Pentzien, S.; Höhm, S.; Rosenfeld, A.; Krüger J. Femtosecond laser-induced periodic surface structures on steel and titanium alloy for tribological applications. Phys. A 2014, 117, 103–110. DOI: 10.1007/s00339-014-8229-2.
- Kunz, C.; Bonse, J.; Spaltmann, D; Neumann, C.; Turchanin, A.; Bartolomé, J.F.; Müller, F.A.; Gräf, S. Tribological performance of metal-reinforced ceramic composites selectively structured with femtosecond laser-induced periodic surface structures. Surf. Sci. 2020, 499, 143917. DOI: 10.1016/j.apsusc.2019.143917.
- Borodaenko, Y.; Syubaev, S.; Gurbatov, S.; Zhizhchenko, A.; Porfirev, A.; Khonina, S.; Mitsai, E.; Gerasimenko, A.V.; Shevlyagin, A.; Modin, E.; Juodkazis, S.; Gurevich, E.L.; Kuchmizhak, A.A. Deep subwavelength laser-induced periodic surface structures on silicon as a novel multifunctional biosensing platform. ACS App. Mater. Interfaces 2021, 13, 54551–54560. DOI: 10.1021/acsami.1c16249.
- Hamad, S.; Moram, S.S.B.; Yendeti, B.; Podagatlapalli, G.K.; Rao, S.V.S.N.; Pathak, A.P.; Mohiddon, M.A.; Soma, V.R. Femtosecond laser-induced, nanoparticle-embedded periodic surface structures on crystalline silicon for reproducible and multiutility SERS platforms. ACS Omega, 2018, 3, 18420–18432. DOI: 10.1021/acsomega.8b02629.
- Cunha, A.; Elie, A.-M.; Plawinski, L.; Serro, A.P.; Botelho do Rego, A.M.; Almeida, A.; Urdaci, M.C.; Durrieu, M.-C.; Vilar, R. Femtosecond laser surface texturing of titanium as a method to reduce the adhesion of Staphylococcus aureus and biofilm formation. Surf. Sci. 2016, 360, 485–493. DOI: 10.1016/j.apsusc.2015.10.102.
- Epperlein, N.; Menzel, F.; Schwibbert, K.; Koter, R.; Bonse, J.; Sameith, J.; Krüger, J.; Toepel, J. Influence of femtosecond laser produced nanostructures on biofilm growth on steel. Surf. Sci. 2017, 418, 420–424. DOI: 10.1016/j.apsusc.2017.02.174.
Comment 3: Why is it used a multi-layer substrate?
Response 3:
We added the following explanation into the text:
The chromium layer was deposited to improve adhesion of the ChVS layer. In turn, the dielectric silicon oxide sublayer minimizes influence of photoexcited free charge carriers from the silicon substrate on LIPSS formation exactly in the ChVS layer.
(lines 108-111)
Comment 4: In section 2 (methods), the authors do not disclose the laser repetition rate used for the fabrication of LIPSS in the array of “craters”. It is however reported that 1KHz and 2KHz were used for the fabrication of lines.
Response 4:
Indeed, fabrication of the craters was caried out with low repetition rates (1-10 Hz) to precisely control the number of pulses, whereas large area of LIPSS were done at high repetition rate (1-2 kHz). We specially explained it by adding the following text in section 2:
The production of the crater arrays was done with very low repetition rates to ensure the precise control of the laser pulse number. For N below 100 the repetition rate was 1Hz, whereas for N above 100 the repetition rates was 10Hz.
(lines 124-127)
Comment 5: Can the repetition rate have an effect on LIPSS morphology?
Response 5:
Thank you for your comment. Our experiment evidences no effect of the laser pulse repetition rate on the LIPSS, at least if the rate is below 1MHz, since at higher rates the laser-induced processes are not competed before the next laser pulse irradiates the surface. However, in our experiments, we employed much less repetition rate. We added the following brief discussion of the fact in Section 3.1. explained it by adding the following text in section 2:
Our study revealed no effect of the laser pulse repetition rate on the type of created surface structures for the repletion rate below 1 MHz, when the changes introduced by the previous laser pulse have not been relaxed by the time of irradiation by the next one.
(lines 264-266).
Comment 6: Row 117: it is not entirely clear to what energy losses are related to.
Response 6:
We are very indebted to the Reviewer for this remark. We must admit that mentioning losses does not help understanding of the text. In fact, we discussed them to find conversion ratio of the laser pulse energy at the laser exit and the laser pulse energy immediately incident at the sample. In the revised text we totally get the former one away and mentioned the pulse energy at the sample only. We hope this will make reading the paper easier. As result, we removed any losses' mentioning (e.g., in lines 123-124) from the text and corrected energy values through the whole text and in Figures 2 and 3. The fluence values in the previous version of the manuscript were correct and they are left in the text.
Comment 7: Figure 1: is it the movement of the laser a raster scan?
Response 7:
Thank you for your remark. Indeed, the sample was moved with the help of two translators. In the text we described it in more detail as follows:
The irradiation of large areas (5x5 mm2) compared to the LIPPS period and laser beam diameter was carried out in the scanning mode, which was realized via moving the sample by a system of 2 automated mechanical translators.
(lines 130-132)
Comment 8. Laser polarization should be indicated in all SEM images with arrows as it was done for Figure 5.
Response 8:
Thank you for your comment. We indicated laser polarization in Figure 5.
Comment 9. Have the authors measured the depth of the produced “craters”? And of the produced “lines”? Was it considered when producing the large area samples?
Response 9:
Thank you for your comment. Our measurements evidence no additional deepening inside the crater or scanning lines. We mentioned this fact in the following phrases:
The depth of the crater coincides with the depth of the LIPSS (Figure 4).
(lines 256-257)
and
The depth of the formed structure is the LIPSS depth (Figure 5d)
(lines 293-294).
Comment 10. How were the LIPSS periodicities measured? I imagine it was carried out by analyzing Fourier transform images, but it was not mentioned in the paper.
Response 10:
No, the analysis Fourier transform images gives a larger error compared with the direct measurement. Therefore, we analysed AFM profiles and noted this in the text (lines 141- 142): “The LIPSS periods were found as the average of 10 or more periods measured at an analysis of AFM profiles.”
Comment 11. One of the strong points of the papers is that the authors were able to obtain two-dimensional LIPSS with a single processing step, a linear polarization and without using beam shaping or other techniques such as “photon doubling”. I suggest highlighting this and to compare it with recently employed techniques such as…
Response 11:
Thank you for your comment. We added short discussion of hierarchical LIPSS (lines 295-300):
Herein, it should be emphasized that the observed fabrication of the hierarchical LIPSSs does not require laser beam splitting for relief modification [46–48], i.e. using the direct laser interference pattering technique [49], or changing a scanning strategy [50]. Thus, we have implemented a single-stage, single-beam and scalable technique to form hierarchical LIPSSs in As2Se3 thin films. Moreover, to our knowledge, we formed LIPSS on As2Se3 films for the first time.
and references:
- Mastellone, M.; Bellucci, A.; Girolami, M.; Serpente, V.; Polini, R.; Orlando, S.; Santagata, A.; Sani, E.; Hitzel, F.; Trucchi, D.M. Deep-subwavelength 2D periodic surface nanostructures on diamond by double-pulse femtosecond laser irradiation. Nano Lett. 2021, 21, 4477−4483. DOI: 1021/acs.nanolett.1c01310.
- Fraggelakis, F.; Mincuzzi, G.; Lopez, J.; Manek-Hönninger, I.; Kling, R. Controlling 2D laser nano structuring over large area with double femtosecond pulses. Surf. Sci. 2019, 470, 677–686. DOI: 10.1016/j.apsusc.2018.11.106.
- Fraggelakis, F.; Tsibidis, G.D.; Stratakis, E. Tailoring submicrometer periodic surface structures via ultrashort pulsed direct laser interference patterning. Phys. Rev. B 2021, 103, 054105. DOI: 1103/PhysRevB.103.054105.
- Bonse, J. Quo vadis LIPSS?—Recent and future trends on laser-induced periodic surface structures. Nanomaterials 2020, 10, 1950. DOI: 10.3390/nano10101950.
- Yu, X.; Zhang, Q.; Qi, D.; Tang, S.; Dai, S.; Zhang, P.; Xu, Y.; Shen, X. Femtosecond laser-induced large area of periodic structures on chalcogenide glass via twice laser direct-writing scanning process. Las. Technol. 2020, 124, 105977. DOI: 10.1016/j.optlastec.2019.105977.
We also corrected mentioned errors in lines 29 and 56.

Reviewer 2 Report
This very interesting article describes the process of creating periodic surface structures on arsenic sulfide and arsenic selenide films using femtosecond laser pulses. The paper is well written. I found only one mistake on page 2, line 56 - the end of the sentence is repeated twice.
Motivation and experimental set-up are well described. The results and conclusions are clear and valuable and I have no remarks except one minor issue. On page 3, line 117 it is written that energy losses are 40%. It is not clear if it is an assumption or was measured.
Author Response
Thank you for high estimation of our paper!
We are very indebted to the Reviewer for this remark. We must admit that mentioning losses does not help understanding of the text. In fact, we discussed them to find conversion ratio of the laser pulse energy at the laser exit and the laser pulse energy immediately incident at the sample. In the revised text we totally get the former one away and mentioned the pulse energy at the sample only. We hope this will make reading the paper easier. As result, we removed any losses' mentioning (e.g., in lines 123-124) from the text and corrected energy values through the whole text and in Figures 2 and 3. The fluence values in the previous version of the manuscript were correct and they are left in the text.
Reviewer 3 Report
Authors studied effects of femtosecond laser on As2S3 and As2Se3 thin films and analyzed the material characteristics after processing. The current version of paper is not suitable for publication. The following modifications are needed.
1. In title, morphological features are too limited to cover the contents of this study. It is suggested to delete morphological features and summary the research contents comprehensively.
2. In the Part of 2.1, please apply the schematic diagram of film and substrate to explain the experimental procedure better.
3. The morphology of films with different number of laser pulses and energy is not convincing, please provide the fitting curve for the ablation threshold.
4. Is there a test of chemical composition conducted on the material? At the line of 137, EDX was mentioned. However, it seems that no specific test results in the paper.
5. Is it possible to provide the application and performance of LIPSS on As2S3 and As2Se3 films?
There are numerous grammar and tense issues in the article.
Round 2
Reviewer 1 Report
The authors replied satisfactorily to all the points raised during the review process, therefore I suggest pubblication in present form.
Author Response
Thank you for your kind reply!
Reviewer 3 Report
Thanks for the response. However, it seems to be no answer to the comment 5 ( Is it possible to provide the application and performance of LIPSS on As2S3 and As2Se3 films?). Please supply it.
Minor editing of English language is required.
Author Response
First of all, we truly do apologize for sending you wrong reply to your comments. Sorry!
Now let us give you proper reply.
Let us express our thanks for your comments on our manuscript.
We tried to improve the manuscript text in accordance with the comments (marked with grey colour)
We also tried to improve the manuscript language (correction are marked with yellow colour).
Comment 1: In title, morphological features are too limited to cover the contents of this study. It is suggested to delete morphological features and summary the research contents comprehensively.
Response 1:
Thank you for your comment. We agreed with your supposition and corected the title of the paper in the following way:
Hierarchical surface structures and large-area nanoscale gratings in As2S3 and As2Se3 films irradiated with femtosecond laser pulses
Comment 2: In the Part of 2.1, please apply the schematic diagram of film and substrate to explain the experimental procedure better.
Response 2:
Thank you for your comment. We added Figure 1a. Scheme of the samples deposited on the multilayer substrate. (page 4, line 153).
Comment 3: The morphology of films with different number of laser pulses and energy is not convincing, please provide the fitting curve for the ablation threshold.
Response 3:
In this paper, we mostly focus at LIPSS formation, which demands irradiation by several pulses (in our cases not less than 10). Meanwhile, during the multi-pulse exposure, the physical processes that occur between the pulses can significantly affect the result of the interaction and complicate the analysis, that is why single-pulse craters are usually used for measuring ablation threshold. Thus, since we did not carry out single-pulse irradiation, precise ablation threshold for the As2S3 and As2Se3 could not be found in our experiments. However, comparing the fluences in our experiment with the ablation threshold values found elsewhere evidences that we exceed the ablation threshold. The short discussion is added to the text (lines 189-191):
It is worth noting that for As2S3 the fluence exceeds the ablation threshold, which ranges from 7.2 mJ/cm2 to 24 mJ/cm2 for femtosecond 800 nm laser pulses according to Refs [41,42].
and
Typically, ablation threshold for As2Se3 was reported to be almost twice less than for As2S3 [44,45].
(lines 193-194).
Comment 4: Is there a test of chemical composition conducted on the material? At the line of 137, EDX was mentioned. However, it seems that no specific test results in the paper.
Response 4:
Thank you for this comment. indeed, we carried out EDX analisys of the structures. No evidences of the surface composition variation was found. We added following text (lines 270-272):
EDX analysis of structures did not reveal any change in the composition of the substance comparing with initial one. The EDX results prove the uniform distribution of elements across the as-deposited and irradiated films (see Supplement).
and submitted a special Supplement to the paper.
Comment 5: Is it possible to provide the application and performance of LIPSS on As2S3 and As2Se3 films?
Response 5:
Nowadays, study of LIPSS on As2S3 and As2Se3 films is limited to their fabrication without any applying in practice. There are only several works regarding this topic exists for As2S3, e.g., Refs. [30] and [50] in our manuscript. To our knowledge, we formed LIPSS on As2Se3 films for the first time. The absence of real practical applications of the studied structures does not mean that this is impossible. There are the necessary prerequisites owing to transparency in the near and middle infrared region and periodical relief. Therefore, we wrote in the end of the article (lines 391-394):
Such 2D periodical gratings on the irradiated As2S3 and As2Se3 surfaces may cause optical artificial anisotropy and are of interest for designing elements of polarization optics and photonic integrated circuits for the infrared range.
Of course, additionally studies are required, and we are going to carry them out in the nearest future. These are topics for further articles.
30. Yu, X.; Qi, D.; Wang, H.; Zhang, Y.; Wang, L.; Zhang, Z.; Dai, S.; Shen, X.; Zhang, P.; Xu, Y. In situ and ex-situ physical scenario of the femtosecond laser-induced periodic surface structures. Opt. Express 2019, 27, 10087–10097. DOI:10.1364/OE.27.010087
50. Yu, X.; Zhang, Q.; Qi, D.; Tang, S.; Dai, S.; Zhang, P.; Xu, Y.; Shen, X. Femtosecond laser-induced large area of periodic structures on chalcogenide glass via twice laser direct-writing scanning process. Opt. Las. Technol. 2020, 124, 105977. DOI: 10.1016/j.optlastec.2019.105977.
Again, thank you for your remarks!
